# Health Assessment of Electronic Waste Workers in Chile: Participant Characterization

**DOI:** 10.3390/ijerph16030386

**Published:** 2019-01-29

**Authors:** Karla Yohannessen, Daniela Pinto-Galleguillos, Denisse Parra-Giordano, Amaranta Agost, Macarena Valdés, Lauren M. Smith, Katherine Galen, Aubrey Arain, Felipe Rojas, Richard L. Neitzel, Pablo Ruiz-Rudolph

**Affiliations:** 1Programa de Salud Ambiental, Instituto de Salud Poblacional, Facultad de Medicina, Universidad de Chile, Independencia 939, Independencia, Santiago 8380453, Chile; karlayohannessen@med.uchile.cl (K.Y.); dpintogalleguillos@gmail.com (D.P.-G.); aagost@uc.cl (A.A.); feliperojaszamora@gmail.com (F.R.); 2Departamento de Enfermería, Facultad de Medicina, Universidad de Chile, Independencia 1027, Independencia, Santiago 8380453, Chile; denisseparrag@gmail.com; 3Programa de Epidemiología, Instituto de Salud Poblacional, Facultad de Medicina, Universidad de Chile, Independencia 939, Independencia, Santiago 8380453, Chile; macavaldes@ug.uchile.cl; 4Department of Environmental Health Sciences, School of Public Health, University of Michigan, 1415 Washington Heights, Ann Arbor, MI 48109, USA; laumarie@umich.edu (L.M.S.); katgalen@umich.edu (K.G.); aubreyll@umich.edu (A.A.); rneitzel@umich.edu (R.L.N.)

**Keywords:** electronic waste recycling, occupational health, public health, injuries, stress

## Abstract

Little research has been done to evaluate the occupational health of electronic waste (e-waste) recycling workers in Latin America. The objective of this study was to complete comprehensive health evaluations on e-waste recycling workers in Chile and to compare those that work in informal (i.e., independent) to those that work in formal (i.e., established company) settings. A cross-sectional study in the summer of 2017 recruited 78 informal recycling workers from two cities and 15 formal e-waste recycling workers from a single recycling facility to assess exposures and health outcomes. Participants completed a health questionnaire and underwent a full health assessment. Herein, only health questionnaire data are reported. Participants were primarily male, middle-aged, married with children, and had worked in e-waste recycling for an average of 12 years. Participants generally reported good health status, and their prevalence of chronic diseases was comparable to national rates. Workers frequently reported exposures to several occupational stressors, including noise and insufficient income, as well as other mental health stressors. Occupational injuries were commonly reported and use of safety equipment was low. Only a few significant differences, generally of a rather small magnitude, were found between informal and formal workers. In conclusion, from survey data, we did not identify major risks to health among e-waste workers, and only minor differences between workers in informal and formal settings.

## 1. Introduction

The World Health Organization (WHO) has declared the production of electronic waste (e-waste) to be a growing global environmental health problem [1,2,3,4,5]. In 2005, annual e-waste generation was estimated at 35 million tons. This figure is increasing, and greater production occurs in developed countries, with later transfer to low- and middle-income countries for reuse or recycling [6]. E-waste is defined as comprising discarded, nonworking electronic products—e.g., cell phones, computers, and home appliances— and their components, which are no longer considered useful by the owner or manufacturer [7]. They are a source of hazardous constituents, such as heavy metals, but also valuable materials, including copper, gold, silver, and rare earth metals [1]. 

E-waste recycling occurs in formal and informal settings [2]. Formal settings involve well-established recycling plants specifically designed for recycling electronics, with at least some proper ventilation and protection for the workers laboring on them. Such facilities recycle between 12% and 35% of all e-waste generated in developed countries [7]. Informal recycling, on the other hand, tends to involve independent workers, or small groups of workers or families—sometimes including children—who often use hazardous techniques such as cutting, acid baths, heating/smelting, and open burning of materials, without the use of personal protection equipment or engineering controls [2,8]. Informal recycling occurs more often in developing countries. Generally, formal recycling of e-waste is considered better for workers and the environment when compared to informal recycling [9].

Conditions often present in informal recycling are associated with increased risks to human and environment health, which has heightened awareness in the scientific communities and governmental agencies [1,2,5]. Human exposure to the pollutants of e-waste recycling is not isolated to the occupational setting and can spill into the environment of the general population, which can include vulnerable populations such as children and pregnant women living in or near informal recycling sites [2,3]. These exposures have been linked to a range of health effects, including disrupted thyroid function and cellular expression, adverse neonatal effects, and decreased lung function, among others [3].

Environmental contamination issues associated with e-waste recycling have been described worldwide [2], as have e-waste disposal policy options, e-waste reduction strategies, and, to a lesser degree, human health impacts of e-waste [2]. Studies focusing mainly on informal recyclers have been conducted in middle–lower income countries such as China [10,11], Ghana [12,13], and Vietnam [14], and have subsequently found high levels of endocrine disruptors and heavy metals in samples of hair, urine, and blood [2]. In China, these contaminants have been found at high levels in the surrounding population as well, specifically in samples from human placenta, breast milk, and blood [15,16,17,18]. On the other hand, among formal recyclers in higher income countries like France [19], Sweden [9] and the United States [20], high levels of heavy metals have been reported in blood, urine, skin, and clothing. While studies have investigated exposures for informal e-waste workers from low-income countries in Asia and Africa, little is known about these exposures in middle- and high-income countries outside the United States and Europe. Very little is known in Latin America, with the exception of a study in Uruguay in a pediatric population [21]. Additionally, while several studies have assessed contamination or human health impacts, few have examined the association between exposure and health outcomes, and even fewer have evaluated these associations among both informal and formal e-waste recycling workers.

As of 2017, Chile had a population of over 17 million inhabitants. It is highly urbanized (>87%) and centralized, with more than 41% of the total population living in the capital city of Santiago [22]. The gross domestic product (GDP) has substantially increased in the past fifteen years, reaching US$247,028 billion in 2016 [23], promoted by a neoliberal market economy [24,25]. The model has positioned Chile as one of the most economically developed countries in the region and a member of the Organization for Economic Cooperation and Development (OECD). However, the Chilean economy is also one of the most unequal in the world, with a Gini coefficient of 0.50 [26]. Chile’s accelerated growth has boosted the technological sector and demand for new electronics. This has resulted in the highest per capita production of e-waste in Latin America, at 9.9 kg per person per year [27]. As a consequence, a small, formal recycling industry has emerged, and, since 2016, new e-waste recycling laws have been passed that regulate certification recycling companies and include Extended Producer Responsibility (EPR) requirements [27,28]. 

Despite the presence of legislation addressing e-waste recycling, informal recyclers operate in the country, though the full extent of their operations is unknown. Informal recycling networks usually include collectors, who collect e-waste from residential areas and companies; recyclers, who dismantle e-waste into their constituent parts and recover valuable materials; scrap dealers, who buy raw materials and products from recyclers and collectors; and repair shops, which use parts salvaged by recyclers to repair broken goods to return them to use. The overall objective of this study is to complete comprehensive health evaluations of e-waste recycling workers in Chile and compare those that work informally (i.e., in independent settings) to those that work formally (i.e., in well-established companies). The nature and location of e-waste activities done by workers in Chile, particularly those in the informal sector, is unknown. Therefore, the first challenge was to identify and recruit informal and formal workers willing to participate. This study was conducted shortly after the passage of e-waste disposal regulations in Chile; therefore, our study represents a baseline comparison between formal and informal e-waste recycling workers. 

## 2. Methods

### 2.1. Study Design and Sites

A cross-sectional study was conducted from July to August 2017. Since the overall population of informal workers is unknown, this study utilized convenience sampling of workers from two cities of the country where informal recycling was thought to occur at significant levels. The two cities sampled were Santiago, the largest city in the country, and Temuco, a mid-size city of about 300,000 inhabitants [22] (Figure 1). To recruit informal workers, in a first step, several network of workers were contacted to identify sections of the cities where recycling activities were usually performed. Then, these sections were visited by field personnel in order to contact individuals for recruitment purposes. On the other hand, formal workers were recruited from a single mid-size formal recycling facility. 

### 2.2. Participant Recruitment

In June and July 2017, informal sites were identified across the two cities at e-waste collection sites, repair shops, public fairs, and flea markets. In July and August 2017, each site was visited and potential participants were identified. Potential participants were then screened to confirm that they fell into one of the recycling activities targeted in our study (i.e., collectors, recyclers, repairers, or scrap dealers). For formal workers, a recycling company was contacted, and authorization was requested from the owner to carry out the study. In both settings, research staff explained the study objectives, procedures, and measurements to potential participants in detail, and interested individuals then signed a Spanish-language consent form. 

### 2.3. Health Assessment

Participant data collection in Santiago, Temuco, and at the formal recycling company took place in August 2017. Since e-waste work is related to several different kinds of exposure (e.g., air pollution, noise, stress, injuries) and many potential health outcomes as shown in previous studies [2], the questionnaire and exposure assessment was comprehensive. Table 1 summarizes the full set of procedures participants completed in the study. All participants were asked to complete a questionnaire and health assessment, provide blood and urine samples, undergo air sampling and noise monitoring, and complete a daily activity log. This manuscript will focus only on the results from the questionnaire. Four items from the questionnaire were drawn from Cohen’s perceived stress scale; these items addressed the frequency with which participants: felt unable to control important things in life; were confident about personal ability to handle problems; felt things were going their way; and felt they could not overcome difficulties [29].

Additionally, a subset of participants also provided surface wipe samples in their homes or workplaces, were filmed for later activity analysis, and participated in focus groups designed to explore particular aspects of informal e-waste recycling. These activities will be described elsewhere.

The study protocol was approved by Health Sciences and Behavioral Sciences Institutional Review Board (IRB-HSBS) of the University of Michigan (Study eResearch ID: HUM00114562), and the Ethics Committee for Research in Human Beings of the Faculty of Medicine, University of Chile (approval 101-2017).

### 2.4. Data Analysis

An initial exploratory analysis was performed to identify, and correct, typing errors and missing values. Then, descriptive analysis of the variables was carried out using absolute and relative frequencies, for categorical variables, and measures of central tendency and dispersion, for continuous ones. Differences between sites and types of recyclers were explored using chi-squared test (*χ*^2^) and analysis of variance (ANOVA), with a significance level of *p* < 0.05.

## 3. Results

### 3.1. Sociodemographic and Occupational Characteristics of the Population

From the initial 190 people invited, 93 agreed to participate: 53 informal recyclers in Santiago, 25 informal recyclers in Temuco, and 15 workers at the formal recycling company (Figure 1). In Santiago, participants were mainly clustered around four sites: Peñalolén, Peñaflor, Maipú, and Persa Bío-Bío, with participants from Peñalolén, Peñaflor, and Maipú being mostly collectors and recyclers, and those from Persa Bío-Bío being mostly repairers. In Temuco, participants were clustered around three sites: Feria Pinto, Corcolén Park, and Emmaus, with participants from Feria Pinto and Emmaus being mostly repairers, and those at Corcolén being recyclers. The 15 participants from the formal recycling plant were mostly operators, with some administrative staff, working in all sections of the recycling process (sorting, dismantling, shredding, packaging, shipping). 

Sociodemographic characteristics of the participants are shown in Table 2. Most participants (74%) were male, with an average age of 47 years, and lived in households of approximately four people. About half of the participants were married with children. Their education level included primarily basic (8 years) or medium school (12 years); only 16% reached higher education with college or technical degrees. Informal workers were slightly more likely to be male, and were slightly older than formal workers. E-waste recycling was the main source of income for most participants (83%). More than one-third of participants (37%) reported other sources of income. Monthly income was around US$500, which is above the minimum salary in Chile (i.e. US$445 per month). More than 20% of participants reported income above US$1000. Notably, there was no difference in reported income between formal and informal workers. Overall, participants had low-to-medium income and were similar to workers in other technical jobs in Chile [30].

Work history was evaluated and presented in Table 3. Most participants were currently employed (97%) and working in e-waste related jobs (99%). They had been involved in e-waste activities for approximately 12 years on average, and most (61%) worked 5–8 hours per day in a 6-day work week on average. The most commonly reported e-waste activities were recycling (62%), collecting (34%), repairing (31%), and dealing scrap (25%). Participants identified their primary role in the e-waste processes as recyclers (46%), collectors (42%), repairers (34%), and scrap dealing (10%). Compared to informal workers, formal workers worked in direct recycling activities, had e-waste work as their main source of income, and had been working in this sector for a shorter time (<2 years on average).

### 3.2. Self-Reported Health

Results for self-reported health are shown in Table 4. Overall, most participants (>60%) assessed their health status as good or better, with a small fraction (~30%) reporting fair or poor health. The most commonly reported acute conditions (>20% reporting occasionally or frequently experiencing these conditions) were headache or dizziness, abnormal heart beat, breathing problems, and nausea or stomach ache in the previous two weeks. A small fraction of the participants (~5%) reported more serious symptoms such as blood in urine and blood in stool. Only a fraction of the workers (~30%) sought medical treatment for any these conditions, mostly in the formal health care system.

Regarding chronic conditions, 31% of participants were current smokers, consistent with the national prevalence of smoking in Chile (33.3%) [31]. Most participants did not report chronic diseases. Among those who did, the most common were high blood pressure (26%), diabetes mellitus (14%), and asthma (5.4%). Medication use was common among participants with these conditions. A small fraction of participants (16%) reported that they suffer conditions that limit their work and a surprisingly high amount (22%) reported unintentional weight loss during the past year. Few significant differences were found when comparing formal and informal workers.

### 3.3. Self-Reported Personal and Occupational Stressors

Personal and occupational stressors are reported in Table 5. Slightly more than 40% of participants felt they were sometimes or very often unable to control important matters in their lives and that they could not overcome difficulties. Insufficient income to support themselves and their families was also reported by 59% of participants. However, most workers reported that they decide their work methods (77%), that they have not experienced violence or harassment at work (83%) and that work does not interfere with their family responsibilities or leisure time (62%). Slightly more formal workers reported that income was not sufficient to support the family (*p* = 0.029); however, the difference was fairly small.

Self-reported exposure to noise is shown in Table 6. A majority (59%) of participants reported working in loud noise sometimes or more often than sometimes, with an average duration of 8.8 ± 11 years. Several participants reported difficulty hearing (28%) that had started in adulthood (89%). However, diagnosis of hearing loss by a medical professional was rare among workers (8.6%). Finally, approximately one-third of workers reported experiencing tinnitus after being exposed to loud noise. Informal workers reported having worked in conditions of loud noise for approximately 2.5 times longer (approximately 9 years) on average than formal workers (3.6 years). 

### 3.4. Self-Reported Occupational Injuries

Participants reported an average of 3 ± 7.1 occupational injuries in the past six months (Table 7). Reported injuries included cuts and lacerations (31%), contusions and abrasions (16%), and puncture wounds (8.6%) most frequently and most often occurred to workers’ hands (38%) and feet or lower legs (16%). Most participants did not receive formal medical care to treat their wounds (69%) while a small fraction (14%) received treatment at a hospital or clinic. Most workers did not miss work due to their injuries (76%) but among those who did miss work, 35% lost five or more days. The most frequent activities at the time of injury were dismantling (32%), sorting (16%) and collecting (8.6%) e-waste. 

Most participants reported that tools and e-waste components (59%) or work tasks (53%) were what caused their injuries. Only 17% of participants reported receiving instruction or training on how to prevent injuries (17%). The use of safety equipment was relatively rare. While 57% of workers reported wearing leather/rubber gloves and 50% reported wearing rubber-soled boots or shoes, only about one-third wore safety glasses or a face shield, and only 16% wore a dust mask. Finally, approximately half (51%) reported moderate pain (visual analogous scale, VAS 4.9 ± 2.0) in their hands or wrists after working with e-waste. Slightly higher prevalence (61%) was found for reported muscle soreness from sitting in the same position. When comparing formal and informal workers we found that formal workers reported less injuries that required less medical care of any kind. Formal workers also reported more frequent use of leather/rubber gloves and rubber-soled boots or shoes than informal workers. Formal workers reported significantly lower intensity of muscle soreness but not prevalence of muscle soreness when compared to informal workers.

## 4. Discussion

While a number of studies of informal e-waste recycling workers have been conducted worldwide [2], little is known about this industry in middle- and high-income countries outside United States and Europe. Our study of 93 workers appears to be the first study of e-waste recyclers in Chile, and adds to the very sparse literature on e-waste recycling in Latin America [21]. Informal sector participants in this study worked for more than a decade, on average, suggesting that this industry has emerged relatively recently in Chile. Workers generally reported good overall health status and the prevalence of chronic diseases reported was comparable to national levels. Although workers reported several stressors, insufficient income to support themselves and their families was the most common issue among all participants. The prevalence of injuries was high (an average of three injuries in the past 6 months), and the use of protective equipment was generally low. The most common injuries were cuts and lacerations to the hands that occurred during the dismantling of e-waste products. Beyond visible injuries, participants reported experiencing pain in their hands and muscle soreness often after e-waste recycling work. The majority of workers reported being exposed to high levels of noise at least sometimes. We found few differences between informal and formal recycling workers; formal workers were slightly younger, more likely to work mainly as recyclers, and had less experience working with e-waste than did the informal workers.

Studies of informal e-waste recycling workers have been conducted in Ghana [12,13,32] China [10,11], Vietnam [14], and India [4]. As with our study, these workers were mostly male, except in Vietnam where workers were mostly female; however, workers in other studies were generally less educated, younger, and had a lower income with longer and more unstable working hours than participants in Chile. Workers in Ghana [32] worked in e-waste recycling for less time than Chilean workers. E-waste recycling activities most often reported by informal workers in other countries included collecting, dismantling, and scraping, while in Chile there was a high prevalence of repairing and reselling of e-waste as well. Health effects and behaviors also differed between our sample and those conducted on informal e-waste recycling workers outside of Chile. A study conducted in Ghana [32] suggests that e-waste recycling workers there smoked less in comparison to our sample, but reported more cardiovascular symptoms. The informal workers in our study reported a range of symptoms that is generally consistent with a previous study [4] that showed informal e-waste recycling work is associated with diseases in the skin, stomach, respiratory tract and other organs. Other existing studies did not report on these conditions and their relation to e-waste recycling.

Studies of formal e-waste recycling workers are uncommon. A handful of such studies have been conducted in the United States [20], France [19], and Sweden [9]. Only the Swedish study characterized its population but did not describe their primary recycling activities. Formal e-waste recycling workers in Chile were mainly involved in recycling activities such as sorting, scraping, dismantling, and baling, which were all activities previously described for other studies of formal workers [9,19,20]. None of these studies had a health history evaluation, so no relevant comparison could be made.

Occupational exposure to stressors among informal e-waste recycling workers assessed in Ghana [32], reported moderate to high levels of stress, work in unfavorable physical conditions, as well as violence or harassment in their occupational environment, and insufficient income to support themselves for most participants. Chilean informal e-waste recycling workers did not report substantial exposures to occupational stressors, violence, or harassment. However, our Chilean participants did commonly report insufficient income to support themselves and their families. Informal e-waste recyclers in Ghana [32] reported greater intensity of occupational noise exposure, while Chilean workers reported longer noise exposure times. Difficulty hearing and reported diagnosis of a hearing loss was twice as frequent among Chilean informal e-waste workers as compared to Ghanaians [32]. We did not identify other studies that quantified occupational injuries among e-waste recycling workers. Overall, it appears that exposure to stressors, and their health impacts, may be lower in Chile than in similar studies performed in developing nations. 

There are limitations to our study that may reduce the generalizability of our findings. First, the representativeness of our sample is unknown, as the workforce of Chilean e-waste recyclers is not well understood, especially in the informal sector. We tried to obtain the most representative sample possible by searching for participants in both informal and formal settings and across several cities in Chile. Second, our finding that workers did not report major health problems may simply be a reflection of the healthy worker effect [33]. Third, our cross-sectional assessment of exposures and health status may not accurately capture changes in these factors that have occurred over time, or the long-term average status and can therefore not prove causality. Finally, it is possible that participant responses were subject to social desirability bias, which could partially explain both the low health risks and the differences found between formal and informal.

## 5. Conclusions

Our study is among the first to directly compare exposures and health status among formal and informal e-waste recycling workers anywhere, and the first study of e-waste recyclers in Chile. The results of our questionnaire indicated that e-waste workers in Chile did not present major chronic health effects but rather small effect for injuries and stress, with the latter possibly due to income insecurity. Only a few significant differences, generally of a rather small magnitude, were found between informal and formal workers. 

Electronic waste recycling in Chile seems to differ from other studied areas in the methods of reuse as well as the population that participates in the sector. Further research is needed to fully understand and evaluate the long-term impact of both formal and informal sectors in Chile. However, this research, in combination with previously published literature, lends credence to the idea that e-waste recycling differs significantly by country and in each case should be considered a special context. Therefore, to establish country-specific and appropriate policies and safety measures while simultaneously maintaining economic flexibility for those involved, evaluation and understanding of the context should be completed first. Finally, by providing a baseline assessment of exposures and health conditions among informal and formal e-waste recycling workers, we have established a comparison point for subsequent studies both to evaluate the impacts of the new e-waste recycling law in Chile and to compare it with other contexts and countries over time.

## Figures and Tables

**Figure 1 ijerph-16-00386-f001:**
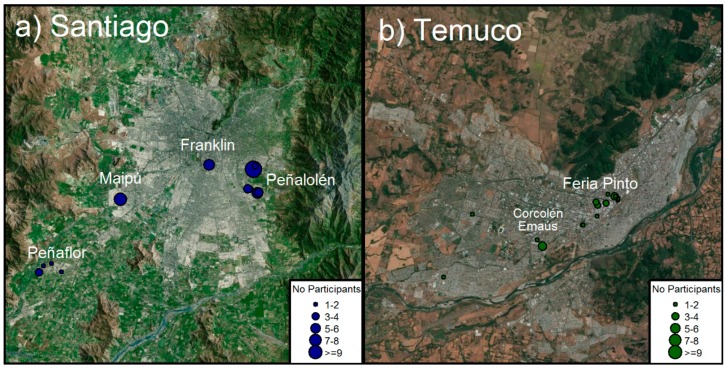
Study site: (**a**) map of Santiago; (**b**) map of Temuco.

**Table 1 ijerph-16-00386-t001:** Summary of participant activities.

Item	Components
Questionnaire	Included: sociodemographic information; work history; self-reported health (including physical and mental health); stress; noise exposure; and occupational injuries (including a Visual Analogous Scale, VAS). Administered by a trained researcher in Spanish (native speaker). The answers were entered using the Qualtrics Software from University of Michigan.
Health assessment	Screening questionnaire smoking habits, current respiratory health, and exclusion criteria for spirometry. Weight and height (Portable Balance *Seca* 813 and Portable Stadiometer *Seca* 213, Hamburg, Germany) Heart rate and blood pressure (electronic sphygmomanometer, Omron, Hoofddorp, Netherlands) Lung-function using forced vital capacity maneuver (Easy One Spirometer, New Diagnostic Design, Andover, MA, USA). Hearing screenings (portable Earscan 3 audiometer, Earscan, Inc, Murphy, NC, USA.). Continuously heart rate during the workday (Polar RS300x watch synced with a Polar H1 heart rate sensing chest strap).
Biomonitoring	Blood samples. Analyzed for lead, cadmium, manganese, aluminum, nickel, iron, zinc (in whole blood), and copper, calcium, and creatinine (in blood serum). Urine samples. Analyzed for lead, cadmium, copper, zinc, manganese, iron, nickel, mercury, aluminum, calcium, and creatinine (NIOSH Method 8310, CDC, USA).
Exposure assessment	Air samples in breathing zone of the workers (AirCheck sample pumps attached to a sampling cassette for the collection of lead according to NIOSH Method 7300). Filters analyzed for: antimony, beryllium, cadmium, chromium, cobalt, copper, iron, lead, manganese, molybdenum, nickel, vanadium, and zinc. Personal noise exposures (Cirrus Dose Badge noise dosimeter, United Kingdom). Surface samples, using a wipe in an area of 10 cm × 10 cm. Analyzed for copper, iron, nickel, manganese, lead, and zinc.
Other procedures	Photographic records and short videos during the work activities. Activity log reporting the amount of time spent in each of worker’s daily activities. Focus groups.

**Table 2 ijerph-16-00386-t002:** Sociodemographic characteristics of the study population, by job type and site.

	All	Informal		Formal	
Characteristics/Groups	*N* = 93	Santiago *N* = 53	Temuco *N* = 25	Recycling Company *N* = 15	*p*-Value ^d^
*Demographic*					
Sex (male), *n* (%)	69 (74)	38 (72)	23 (92)	8 (53)	**0.021**
Age (years), mean ± SD	47 ± 14	47 ± 15	52 ± 12	39 ± 13	**0.018**
Marital status, *n* (%)					
- Married	41 (44)	18 (34)	15 (60)	8 (53)	0.099
- Single	37 (40)	27 (51)	5 (20)	5 (33)	
- Living with partner	8 (8.6)	4 (7.5)	4 (16)	0 (0)	
- Separated	4 (4.3)	1 (1.9)	1 (4)	1 (13)	
- Divorced	2 (2.2)	2 (3.8)	0 (0)	0 (0)	
- Widowed	1 (1.1)	1 (1.9)	0 (0)	0 (0)	
Family members ^a^, mean ± SD	3.8 ± 1.8	3.8 ± 1.8	3.8 ± 1.9	3.5 ± 1.8	0.774
Children <21 years old, *n* (%)	45 (48)	28 (53)	11 (44)	6 (40)	0.596
Level of school, *n* (%)					
- None	7 (7.5)	5 (9.4)	2 (8)	0 (0)	0.336
- Basic school (8 years)	30 (32)	21 (40)	6 (24)	3 (20)	
- Medium school (12 years)	41 (44)	18 (34)	14 (56)	9 (60)	
- College or higher	15 (16)	9 (17)	3 (12)	3 (20)	
*Socioeconomic*					
Main sources of income ^b^, *n* (%)					
- Electric or electronic waste	77 (83)	45 (85)	20 (80)	12 (80)	0.672
- Other types of waste	34 (37)	29 (55)	5 (20)	0 (0)	**0.000**
- Other	20 (22)	12 (23)	6 (24)	2 (13)	0.178
Monthly income ^c^, *n* (%)					
- Less than $165	4 (4.3)	4 (7.5)	0 (0)	0 (0)	0.527
- $167–495	28 (30)	19 (36)	6 (24)	3 (20)	
- $497–1156	35 (38)	19 (36)	9 (36)	7 (47)	
- $1156–1651	15 (16)	4 (7.5)	8 (32)	3 (20)	
- More than $1651	5 (5.4)	2 (3.8)	2 (8)	1 (6.7)	
- Prefer not to answer	6 (6.5)	5 (9.4)	0 (0)	1 (6.7)	

^a^ Supported by household income; ^b^ There could be more than one source of income; ^c^ Parity of chilean peso (CLP) $606 per dollar at 04 April 2018; ^d^ chi^2^ for the comparison between formal and informal and categories in the variable or ANOVA for quantitative variables; In bold significant values (*p* < 0.05).

**Table 3 ijerph-16-00386-t003:** Work history and recycling activities, by job type and site.

	All	Informal		Formal	
	*N* = 93	Santiago *N* = 53	Temuco *N* = 25	Recycling Company *N* = 15	*p*-Value ^b^
Currently employed, *n* (%)	90 (97)	50 (94)	25 (100)	15 (100)	0.310
Currently involved in e-waste activities, *n* (%)	92 (99)	52 (98)	25 (100)	15 (100)	0.683
Years working with e-waste, mean ± SD	12 ± 12	14 ± 12	16 ± 12	2.3 ± 3.2	**0.001**
Hours of work per day, *n* (%)					
- Less than 5 hours	4 (4.3)	3 (5.8)	1 (4.0)	0 (0)	0.053
- 5–8 hours	56 (61)	26 (50)	16 (64)	14 (93)	
- More than 8 hours	32 (34)	23 (44)	8 (32)	1 (6.7)	
Work days per week, mean ± SD	5.8 ± 1.0	5.7 ± 1.3	5.9 ± 0.6	5.8 ± 0.3	0.763
Current job(s) ^a^, *n* (%)					
- Electric or electronic waste recycling	58 (62)	36 (68)	8 (32)	14 (93)	**0.000**
- Electronics collection	32 (34)	26 (49)	6 (24)	0 (0)	**0.001**
- Electronics repairer	29 (31)	14 (26)	15 (60)	0 (0)	**0.000**
- Scrap dealer	23 (25)	21 (40)	2 (8)	0 (0)	**0.001**
- Retired	5 (5.4)	4 (7.5)	1 (4)	0 (0)	0.488
- Other	28 (30)	17 (32)	8 (32)	3 (20)	0.648
Role in e-waste recycling, *n* (%)					
- Recyclers	43 (46)	29 (55)	3 (12)	11 (73)	**0.000**
- Collectors	39 (42)	35 (66)	4 (16)	0 (0)	**0.000**
- Repair shop	32 (34)	14 (26)	18 (72)	0 (0)	**0.000**
- Raw materials buyer	9 (9.7)	6 (11)	0 (0)	3 (20)	0.097

^a^ Multiple choices allowed; ^b^ chi^2^ for the comparison between formal and informal and categories in the variable or ANOVA for quantitative variables; In bold significant values (*p* < 0.05).

**Table 4 ijerph-16-00386-t004:** Health characteristics of the study population, by job type and site.

	All	Informal		Formal	
	*N* = 93	Santiago *N* = 53	Temuco *N* = 25	Recycling Company *N* = 15	*p*-Value ^d^
Overall health, *n* (%)					
- Excellent	3 (3.2)	2 (3.8)	1 (4)	0 (0)	0.701
- Very good	10 (11)	5 (9.4)	3 (12)	2 (13)	
- Good	45 (48)	23 (43)	13 (52)	9 (60)	
- Fair	30 (32)	18 (34)	8 (32)	4 (27)	
- Poor	5 (5.4)	5 (9.4)	0 (0)	0 (0)	
Symptoms in last two weeks (occasionally, always or frequently ^a^), *n* (%)					
- Headache or dizziness	44 (47)	27 (51)	12 (48)	5 (33)	0.674
- Heart beating abnormally	28 (30)	24 (45)	3 (12)	1 (6.7)	**0.008**
- Breathing problems	23 (25)	17 (32)	4 (16)	2 (13)	0.288
- Nausea or stomach ache	22 (24)	13 (25)	6 (24)	3 (20)	0.962
- Skin rashes	15 (16)	8 (15)	5 (20)	2 (13)	0.842
- Loose or watery stools	14 (15)	9 (17)	3 (12)	2 (13)	0.408
- Fever	9 (9.7)	6 (11)	3 (12)	0 (0)	0.625
- Shaking or tremors	6 (6.5)	4 (7.6)	2 (8)	0 (0)	0.741
- Blood in urine	3 (3.3)	3 (5.7)	0 (0)	0 (0)	0.674
- Blood in stool	2 (2.2)	1 (1.9)	1 (4)	0 (0)	0.479
Sought medical care/treatment ^b^, *n* (%)	27 (29)	18 (34)	5 (20)	4 (27)	0.632
Type of medical care ^c^, *n* (%)					
- Clinic/hospital	25 (93)	18 (100)	4 (80)	3 (75)	0.080
- Traditional medicine	1 (3.5)	0 (0)	1 (20)	0 (0)	
- Other	1 (3.5)	0 (0)	0 (0)	1 (25)	
Current smoker, *n* (%)	29 (31)	20 (38)	6 (24)	3 (20)	0.281
Chronic diseases, *n* (%)					
- None	47 (51)	22 (42)	14 (56)	11 (73)	0.076
- High blood pressure	24 (26)	14 (26)	8 (32)	2 (13)	0.421
- Diabetes mellitus	13 (14)	8 (15)	5 (20)	0 (0)	0.197
- Asthma	5 (5.4)	3 (5.7)	1 (4)	1 (6.7)	0.927
- Heart disease	5 (5.4)	5 (9.4)	0 (0)	0 (0)	0.136
- Stroke	4 (4.3)	2 (3.8)	1 (4)	1 (6.7)	0.885
- Kidney disease	3 (3.2)	1 (1.9)	1 (4)	1 (6.7)	0.631
- Liver disease	1 (1.1)	0 (0)	1 (4)	0 (0)	0.253
- Other	18 (19)	16 (30)	2 (8)	0 (0)	**0.008**
Taking medication for any of these conditions ^c^, *n* (%)	29 (63)	20 (65)	6 (55)	3 (75)	0.735
Health problems that limit work, *n* (%)	15 (16)	15 (28)	0 (0)	0 (0)	**0.001**
Unintentional weight loss last year, *n* (%)	21 (23)	13 (25)	2 (8)	6 (40)	0.056

^a^ Allowed options were “rarely or never”, “occasionally”, and “always or frequently” (more details in Appendix A). ^b^ One participant answered "Don’t know" (1.1%); ^c^ the percentage reflects the participants that answered “yes” to the previous question; ^d^ chi^2^ for the comparison between formal and informal and categories in the variable; In bold significant values (*p* < 0.05).

**Table 5 ijerph-16-00386-t005:** Self-reported personal and occupational stressors in the study population, by job type and site.

	All	Informal		Formal	
	*N* = 93	Santiago *N* = 53	Temuco *N* = 25	Recycling Company *N* = 15	*p*-Value ^c^
**Stressors by Cohen’s perceived stress scale**					
In the last month, how often have you felt (sometimes, fairly often or very often ^a^), *n* (%):					
- Unable to control important things in life	39 (42)	25 (47)	10 (40)	4 (27)	0.388
- Confident about personal ability to handle problems	85 (92)	47 (89)	23 (92)	15 (99)	0.284
- Things were going your way	85 (92)	47 (90)	24 (96)	14 (93)	0.301
- That you couldn’t overcome difficulties	38 (41)	25 (47)	7 (28)	6 (40)	0.619
**Other stressors** (occasionally, always or frequently, ^a^), n (%)					
- Someone else deciding work methods/pace/order	21 (23)	9 (17)	8 (32)	4 (27)	0.359
- Experiencing violence or harassment at work ^b^	15 (16)	8 (15)	7 (28)	0 (0)	0.066
- Work interfering with family responsibilities/leisure time	35 (38)	21 (40)	10 (40)	4 (27)	0.880
- Income not sufficient to support family	55 (59)	37 (70)	9 (36)	9 (60)	**0.029**

^a^ Allowed options were: never, almost never, sometimes, fairly often and very often; or “rarely or never”, “occasionally”, “always or frequently” (more details in Appendix A); ^b^ One participant answered "Prefer not to answer" (1 (1.1%)); ^c^ chi^2^ for the comparison between formal and informal and categories in the variable; In bold significant values (*p* < 0.05).

**Table 6 ijerph-16-00386-t006:** Self-reported noise exposure in the study population, by job type and site.

	All	Informal		Formal	
Group	*N* = 93	Santiago *N* = 53	Temuco *N* = 25	Recycling Company *N* = 15	*p*-Value ^b^
Exposed to loud noise at work, *n* (%)					
- Never	21 (23)	12 (23)	7 (28)	2 (13)	0.088
- Almost never	17 (18)	10 (19)	6 (24)	1 (6.7)	
- Sometimes	26 (28)	20 (38)	3 (12)	3 (20)	
- Fairly often	16 (17)	5 (9.4)	6 (24)	5 (33)	
- Very often	13 (14)	6 (11)	3 (12)	4 (27)	
Years working in loud noise, mean ± SD	8.8 ± 11	7.8 ± 9.7	16 ± 14	3.6 ± 6.7	**0.004**
Experienced difficulties hearing, *n* (%)	26 (28)	12 (23)	9 (36)	5 (33)	0.414
Time with difficulties hearing ^a^, *n* (%)					
- Since childhood	2 (7.7)	1 (8.3)	0 (0)	1 (20)	0.548
- Since adolescence	1 (3.8)	1 (8.3)	0 (0)	0 (0)	
- Since adulthood	23 (89)	10 (83)	9 (100)	4 (80)	
Diagnosed with hearing loss, *n* (%)	8 (8.6)	4 (7.5)	2 (8)	2 (13)	0.774
Experienced tinnitus after spending time in loud noise, *n* (%)					
- Never	48 (52)	27 (51)	15 (60)	6 (40)	0.255
- Almost never	14 (15)	6 (11)	4 (16)	4 (27)	
- Sometimes	25 (27)	17 (32)	6 (24)	2 (13)	
- Fairly often	4 (4.3)	2 (3.8)	0 (0)	2 (13)	
- Very often	2 (2.2)	1 (1.9)	0 (0)	1 (6.7)	

^a^ percentage reflects the participants that answered “yes” to the previous question; ^b^ chi^2^ for the comparison between formal and informal and categories in the variable or ANOVA for quantitative variables; In bold significant values (*p* < 0.05).

**Table 7 ijerph-16-00386-t007:** Self-report injuries in the study population, by job type and site.

	All	Informal		Formal	
Group	*N* = 93	Santiago *N* = 53	Temuco *N* = 25	Recycling Company *N* = 15	*p*-Value ^c^
Injuries in e-waste recycling work in past 6 months, mean ± SD	3.0 ± 7.1	2.2 ± 3.4	5.4 ± 13	1.7 ± 2.0	0.131
*For the worst injury during e-waste recycling work:*					
Type of injury ^a^, *n* (%)					
- Cuts/lacerations	29 (31)	20 (38)	7 (28)	2 (13)	0.182
- Contusions/abrasions	15 (16)	7 (13)	4 (16)	4 (27)	0.457
- Punctured wounds	8 (8.6)	8 (15)	0 (0)	0 (0)	**0.037**
- Sprains/strains	4 (4.3)	1 (1.9)	2 (8)	1 (6.7)	0.410
- Burns/scalds	3 (3.2)	0 (0)	3 (12)	0 (0)	**0.015**
- Fractures	1 (1.1)	1 (1.9)	0 (0)	0 (0)	0.683
- Other	18 (19)	7 (13)	7 (28)	4 (27)	0.224
Body part(s) injured ^a^, *n* (%)					
- Hand	35 (38)	22 (42)	12 (48)	1 (6.7)	**0.022**
- Foot/Lower leg	15 (16)	11 (21)	2 (8)	2 (13)	0.564
- Hip	4 (4.3)	2 (3.8)	1 (4)	1 (6.7)	0.885
- Other	18 (19)	6 (11)	7 (28)	5 (33)	0.072
Medical care received ^a^, *n* (%)					
- Self-administered first aid	28 (40)	22 (60)	5 (23)	1 (9.1)	**0.007**
- No medical care	20 (29)	7 (19)	10 (46)	3 (27)	
- Treatment at hospital/clinic	10 (14)	4 (11)	2 (9.1)	4 (36)	
- Other	12 (17)	4 (11)	5 (23)	3 (27)	
Missed work due to injuries ^a^, *n* (%)					
- Did not miss any work and worked regular job	53 (76)	30 (81)	17 (77)	6 (55)	0.070
- Did not miss any work and could not do regular job	3 (4.3)	3 (8.1)	0 (0)	0 (0)	
- Missed work	14 (20)	4 (11)	5 (23)	5 (46)	
Working days lost ^a^, n (%)					
- Less than 1 day	1 (7.1)	0 (0)	1 (20)	0 (0)	0.100
- 1–5 days	3 (21)	0 (0)	1 (20)	2 (40)	
- 5–7 days	3 (21)	2 (50)	0 (0)	1 (20)	
- More than 7 days	2 (14)	2 (50)	0 (0)	0 (0)	
Activity at the time of injury ^a^, *n* (%)					
- Dismantling electronic equipment	30 (32)	17 (32)	9 (36)	4 (27)	0.829
- Sorting electronic waste	15 (16)	7 (13)	6 (24)	2 (13)	0.457
- Collecting electronic waste	8 (8.6)	6 (11)	1 (4)	1 (6.7)	0.537
- Removing covering of wires	2 (2.2)	2 (3.8)	0 (0)	0 (0)	0.462
- Burning activities	0 (0)	0 (0)	0 (0)	0 (0)	-
- Ash/wire collection after burning	0 (0)	0 (0)	0 (0)	0 (0)	-
- Other	24 (26)	12 (23)	8 (32)	4 (27)	0.676
Tools/parts of work lead to more frequent injuries, *n* (%)	55 (59)	37 (70)	14 (56)	4 (27)	**0.031**
Reported job tasks that have led to more injuries, *n* (%)	49 (53)	29 (55)	14 (56)	6 (40)	0.417
Report instructions/training prior to injury, *n* (%)	12 (17)	8 (22)	3 (14)	1 (9.1)	0.510
Use of safety equipment at work, *n* (%)					
- Leather/rubber globes	53 (57)	32 (60)	8 (32)	13 (87)	**0.002**
- Rubber-soled boots or shoes	46 (50)	17 (32)	15 (60)	14 (93)	**0.001**
- Safety glasses/face shields/eye protection	33 (36)	21 (40)	9 (36)	3 (20)	0.373
- Dust mask	15 (16)	8 (15)	4 (16)	3 (20)	0.901
- Latex/plastic gloves	11 (12)	6 (11)	5 (20)	0 (0)	0.163
- Earplugs or earmuffs	8 (8.6)	4 (7.5)	2 (8)	2 (13)	0.774
- Other	18 (19)	10 (19)	4 (16)	4 (27)	0.704
Pain in hands/wrists after e-waste working, *n* (%)	47 (51)	27 (51)	11 (44)	9 (60)	0.778
Intensity of pain in hands/wrists (VAS 0–10) ^b^, mean ± SD	4.9 ± 2.0	5.0 ± 2.1	4.9 ± 1.8	4.6 ± 1.8	0.913
Muscle soreness from sitting in the same position, *n* (%)	57 (61)	36 (68)	12 (48)	9 (60)	0.260
Intensity of muscle soreness (VAS 0–10) ^b^, mean ± SD	5.0 ± 2.2	5.4 ± 2.1	4.8 ± 2.1	3.4 ± 1.7	**0.044**

VAS: visual analogous scale; ^a^ percentages reflect participants that reported injuries on the first question; ^b^ reflects the participants that answered “yes” to the previous question; ^c^ chi^2^ for the comparison between formal and informal and categories in the variable or ANOVA for quantitative variables; In bold significant values (*p* < 0.05).

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
