# Peer review of "Health Assessment of Electronic Waste Workers in Chile: Participant Characterization"

_ijerph, 2019, doi:10.3390/ijerph16030386_

Reviewer 1 Report

Title

needs refinement for clarity. I am not sure that what the title reflects is discussed in the abstract. Why state “study design”? is this paper describing baseline measurements of a longer study? Is this a methods paper? The title needs to agree with what is summarized and stated in the abstract. You could consider a more general title to avoid confusing readers: 

Qualitative health assessment of electronic waste workers in Chile

OR

Self-reported health assessment baseline of electronic waste workers in Chile

Or similar

Abstract

The abstract needs considerable work (as the rest of the paper). 

Starting of the abstract: The first two phrases could be written better. I disagree that there is little research on health in informal e-waste recycling as there are hundreds of papers. Perhaps there is a need for longitudinal and robust health studies, but studies are many. There are many gaps, so if you want to refer to that then you need to be very specific. What “health” outcomes are you referring to specifically? You could start the abstract with some factual things or gaps on the health of e-waste recyclers. Would be most ideal if you just start saying that no studies in this industry have been done in Chile, be specific.

End of the abstract: here authors go back and forth between informal and formal and it is very confusing. You say you found the same in both informal and formal yet you assert that informal has challenges, why? The abstract should stand alone and a reader should not be expected to have to read the paper to understand the complete message. Be explicit if you do want to emphasize informal faces challenges even if your data does not sustain that statement per se.

Introduction

You need more specific better defined objectives explaining your study. Without clear objectives, it would be hard to understand the rest of the paper.

You spend too much justifying Chile as a developed country, while not stating any numbers on what you know about this industry nationwide in Chile. What are the stats about how many employees are working in this sector? You have such small study that you need to talk about the size of this industry so the reader has an idea of how representative your sample is of the country or the area where you choose the study. 

It is extremely confusing to read back and forth about informal and formal. It needs to be written very artfully to be able to be compassing without confusing the reader.

You state what the overall objective of the study is but you never say how you will do so (what is the actual study design?) or overall hypothesis? is it meant to be a longitudinal study? Cross sectional only? Health outcomes to be studied? Why are you doing all those measurements?

Methods

Participant recruitment

You need to be more specific how you got to these workers. If you are comparing formal and informal, why different number of employees in both groups? is that a fair comparison? if comparing informal and formal was your main objective why so few formal employees? All formal workers are from the same facility, can you assume that is representative enough so that you can compare with informal without bias? What was your power calculation? Will you see differences once you start getting quantitative data to compare formal and informal or you plan to lump everyone together given the little you recruited in formal? What are you overarching hypothesis/health outcomes of the overall study?

Overall comment 

Studying the e-waste recycling in Chile is gravely needed and a worthwhile effort.

This study should be published and I commend the teams efforts. However, this paper ambitiously wants to be a methods paper with some baseline qualitative data, which prove hard for the authors to write in a clear way. From title, abstract and introduction is hard to decipher clearly what the intention of the paper is. In methods and results in unclear wha the overall study is about. When reading the paper one gets confused between informal and formal e-waste recycling throughout. Baseline data is nice to see in results but not clearly defined in objectives. 

If authors want this to be a methods paper to refer future publications then the methodology of the larger study is unclear as it lacks overall hypothesis, or even study design (is this a longitudinal study or cross sectional or else)? You talk about “study design” but you never state what is the study design that I was able to find. What are the health outcomes that you are studying? What is the power you need to answer your hypothesis? why do biomonitoring? why the other measurements? if sample size was small because it was designed for example for biomonitoring, which is expensive, then data on biomonitoring should be included along with self-disclosed health data.

The authors should ideally use an editing service to check the paper English for clarity and technical style. Editing of the paper is needed to avoid excess words, checking punctuation, and ease of reading. 

Better organization of the paper is also needed for clarity, especially as it is trying to include both a methods paper and baseline information into one paper. Not sure if maybe making the study information a supplemental information so that the baseline data takes front stage and it is easier to digest stepwise. Authors should consider focusing on baseline data and only talking about the overall study briefly, or considering adding more quantitative data to strengthen the paper (answering clear hypothesis) and only mentioning the parts of the study relevant to the data presented to avoid confusion (or write both overall study methods and data in a way that is very clear to the reader). Focusing onto baseline measurements would be important as much is to be discussed to be able to include self disclose measurements and make sense of them in the overall context of Chile health.

Author Response

Response to Reviewer 1 Comments

Point 1: Title: needs refinement for clarity. I am not sure that what the title reflects is discussed in the abstract. Why state “study design”? is this paper describing baseline measurements of a longer study? Is this a methods paper? The title needs to agree with what is summarized and stated in the abstract. You could consider a more general title to avoid confusing readers:

Qualitative health assessment of electronic waste workers in Chile

OR

Self-reported health assessment baseline of electronic waste workers in Chile

Or similar

Response 1: We considered this point on the title along with the overall point about the scope of the paper. We agree that the title was misleading and that the focus of the paper lacked clarity. We have adopted a new, more “narrow” title: “Health assessment of electronic waste workers in Chile: participant characterization”

Point 2: Abstract. The abstract needs considerable work (as the rest of the paper).

Starting of the abstract: The first two phrases could be written better. I disagree that there is little research on health in informal e-waste recycling as there are hundreds of papers. Perhaps there is a need for longitudinal and robust health studies, but studies are many. There are many gaps, so if you want to refer to that then you need to be very specific. What “health” outcomes are you referring to specifically? You could start the abstract with some factual things or gaps on the health of e-waste recyclers. Would be most ideal if you just start saying that no studies in this industry have been done in Chile, be specific.

Response 2: We agree that the first few sentences in the abstract were misleading. We rewrote them giving a clear focus on contrasting the amount of work done on the topic in our region compared to the rest of the world. Then, many parts of the abstract were rewritten for clarity with no major changes in meaning. We did declare more clearly an objective, a study design (cross-sectional), conclusions, and we made clear we are only reporting the questionnaire in this article.  Regarding the points on formal vs. informal, we clearly stated this in the objective, and modified many sections of the paper (including tables) to better assess the differences. Also, we described more in detail what is understood by formal and informal in the introduction but here (in the abstract) we anticipated informal were “independent” workers while formals worked in well-established companies. Finally, regarding the reviewer’s comment about the extensive literature on e-waste recycling, that is correct; there are many papers on this topic in general.  However, many of these papers focus on policy concepts, waste disposal processes, and other topics that are not so relevant to the current study. Many studies have assessed environmental contamination without connecting it to human health. There are far fewer papers directly assessing health impacts, and fewer still that have sought to evaluate the association between health impacts and exposures. Ours appears to be the first to directly compare informal and formal workers as mentioned before. 

Point 3: End of the abstract: here authors go back and forth between informal and formal and it is very confusing. You say you found the same in both informal and formal yet you assert that informal has challenges, why? The abstract should stand alone and a reader should not be expected to have to read the paper to understand the complete message. Be explicit if you do want to emphasize informal faces challenges even if your data does not sustain that statement per se.

Response 3: We agree with the reviewer. The end of the abstract was modified giving a more straightforward conclusion that also agrees with the conclusion in the discussion section. Now it reads: “In conclusion, from survey data, we did not identify major risks to health among e-waste workers, and only minor differences between informal and formals.”

Point 4: Introduction. You need more specific better defined objectives explaining your study. Without clear objectives, it would be hard to understand the rest of the paper.

Response 4: As mentioned before we better defined the scope of the paper and, consequently, we rewrote the last paragraph of the introduction giving a clearer view of the objective. We also removed mentions to the other parts of the larger project, so to give more clarity to the reader.

Point 5: You spend too much justifying Chile as a developed country, while not stating any numbers on what you know about this industry nationwide in Chile. What are the stats about how many employees are working in this sector? You have such small study that you need to talk about the size of this industry so the reader has an idea of how representative your sample is of the country or the area where you choose the study.

Response 5: This is an interesting point that we would like to address.

i)                   We try to give a better context of the study and background on economy in Chile because the work in electronic recycling can vary between countries of different contexts, and for that reason we consider it necessary to make known the Chilean context.

ii)                 Because of i) we know Chile is producing lots of e-waste and the intro give some numbers about e-waste production. Also, a small, formal recycling industry has emerged. This was included in the text.

iii)               Regarding informal e-waste workers in Chile, the numbers are really absent. We just do not have figures of what is going on, so we actually highlight this in the last paragraph of the intro stating that just identifying and studying them was a challenge.

We hope that these clarifications help the reviewer understand our logic.

Point 6: It is extremely confusing to read back and forth about informal and formal. It needs to be written very artfully to be able to be compassing without confusing the reader.

Response 6: This is a crucial point for our paper and we are grateful the reviewer made us see it. We clearly stated this in the objective, and modified many sections of the paper (including tables) to better assess the differences We rewrote this topic completely in a new second paragraph of the intro (“E-waste recycling can occur in formal and informal settings...”), also adding sections regarding this topic in several parts of the paper.

Point 7: You state what the overall objective of the study is but you never say how you will do so (what is the actual study design?) or overall hypothesis? is it meant to be a longitudinal study? Cross sectional only? Health outcomes to be studied? Why are you doing all those measurements?

Response 7: We appreciate the reviewer comments. Many things related to objectives, design and outcomes were changed in the abstract, intro and methods. This was intended to be a descriptive study, without a formal hypothesis statement, although it is understood that in this type of work we hypothesize that e-waste workers can be exposed to many stressors, and hence have health impacts.  We explored how these exposures may vary between formal and informal settings because of the different work conditions.

Point 8: Methods, Participant recruitment

You need to be more specific how you got to these workers. If you are comparing formal and informal, why different number of employees in both groups? is that a fair comparison? if comparing informal and formal was your main objective why so few formal employees? All formal workers are from the same facility, can you assume that is representative enough so that you can compare with informal without bias? What was your power calculation? Will you see differences once you start getting quantitative data to compare formal and informal or you plan to lump everyone together given the little you recruited in formal? What are you overarching hypothesis/health outcomes of the overall study?

Response 8: Again, this is a very interesting point. We rewrote the first sentence on methods more clearly stating the design of the study and the approach to participant selection. We made clear the exploratory nature of the study as the population of informal e-waste workers is unknown. In this context we value the notion of learning about this so poorly characterized group and compare them (although in a narrow way) to their formal counterparts. We declare this also in the discussion.  No power calculations were performed, and no formal hypotheses stated, for this initial, descriptive study.

Point 9: Overall comment, Studying the e-waste recycling in Chile is gravely needed and a worthwhile effort. This study should be published and I commend the team efforts. However, this paper ambitiously wants to be a methods paper with some baseline qualitative data, which prove hard for the authors to write in a clear way. From title, abstract and introduction is hard to decipher clearly what the intention of the paper is. In methods and results in unclear what the overall study is about. When reading the paper one gets confused between informal and formal e-waste recycling throughout. Baseline data is nice to see in results but not clearly defined in objectives.

Response 9: We greatly value that the reviewer appreciates our work and efforts. We think that most of these comments have been already answered. Although if the paper is still unclear or confusing, we would like to know so we can improve it.

Point 10: If authors want this to be a methods paper to refer future publications then the methodology of the larger study is unclear as it lacks overall hypothesis, or even study design (is this a longitudinal study or cross sectional or else)? You talk about “study design” but you never state what is the study design that I was able to find.

Response 10: We included a design of the study and removed the word design from title. We also gave a clearer focus in abstract, introduction and objectives.

Point 11: What are the health outcomes that you are studying? What is the power you need to answer your hypothesis? why do biomonitoring? why the other measurements? if sample size was small because it was designed for example for biomonitoring, which is expensive, then data on biomonitoring should be included along with self-disclosed health data.

Response 11: with regard to the selection of outcomes, we added a sentence in methods clarifying the point: “Given the nature of the e-waste activities relevant exposures are related to environmental pollutants, noise, stressors, injuries, and health endpoints that might be affected, as shown in previous studies”. As there were many outcomes involved, we did not carry out a formal sample size calculation but estimated a size based on previous studies and funding available. Regarding biomonitoring data, we expect to write other papers analyzing in detail other variables and relationships between them.

Point 12: The authors should ideally use an editing service to check the paper English for clarity and technical style. Editing of the paper is needed to avoid excess words, checking punctuation, and ease of reading.

Response 12:  We extensively revised the writing and had two native English-speaking authors go through the text line by line. We hope this current version is suitable for an international audience.

Point 13: Better organization of the paper is also needed for clarity, especially as it is trying to include both a methods paper and baseline information into one paper. Not sure if maybe making the study information a supplemental information so that the baseline data takes front stage and it is easier to digest stepwise. Authors should consider focusing on baseline data and only talking about the overall study briefly, or considering adding more quantitative data to strengthen the paper (answering clear hypothesis) and only mentioning the parts of the study relevant to the data presented to avoid confusion (or write both overall study methods and data in a way that is very clear to the reader). Focusing onto baseline measurements would be important as much is to be discussed to be able to include self-disclose measurements and make sense of them in the overall context of Chile health.

Response 13: we agree with many of these comments. As there are many aspects involved, and many already covered, in this point, our answer is that we expect that all structural changes taken so far would answer most of these remarks.

Reviewer 2 Report

Dear authors,

The social impact related to WEEE management is an important issue asking for valuable contributions from both the scientific and industrial community. Considering this work, your focus goes in the right direction, but the paper requires a strong improvement before publishing.

1) By reading your work, I cannot comprehend were is the innovation of this work. Already from the abstract what remain impressed in mind is that there are no differences between common workers and WEEE workers. So, why we need to study this phenomenon?

2) Section 1 must be revised. In its current state, it seems the sum of independent paragraphs.

3) Section 2 must be revised. A better explanation in terms of procedures followed for the development of both the questionnaire and the interviews must be described within the text.

4) Section 4 must be revised. In its current state, it seems the sum of independent paragraphs.

5) No conclusions are reported. Please, consider a dedicated section. 

Author Response

Response to Reviewer 2 Comments

The social impact related to WEEE management is an important issue asking for valuable contributions from both the scientific and industrial community. Considering this work, your focus goes in the right direction, but the paper requires a strong improvement before publishing.

Point 1: 1) By reading your work, I cannot comprehend were is the innovation of this work. Already from the abstract what remain impressed in mind is that there are no differences between common workers and WEEE workers. So, why we need to study this phenomenon?

Response 1: We appreciate the reviewer comments. We have changed many things related to objectives, design and outcomes in the abstract, introduction and methods so to better convey our logic and the relevance of the paper. According to the literature there are differences between working in recycling of electronic waste in a formal or informal way. Studies with formal workers have been conducted mainly in high-income countries (Europe and US) and informal workers have been studied mainly in low-middle-income countries (Asia, Africa). Our research is carried out in a country recently declared as high income, however there are still many gaps in occupational health issues of middle-income countries. For this reason, we think it is interesting to explore how the exposures may vary between formal and informal settings and we have added to the tables the comparisons between the groups of workers according to site and type of work (before in supplementary material).

Point 2: 2) Section 1 must be revised. In its current state, it seems the sum of independent paragraphs.

Response 2: We agree with the reviewer. We have reviewed and corrected the section 1 for better understanding. In general, the introduction was extensively rewritten and we hope now it is more satisfactory for a reader.

Point 3: 3) Section 2 must be revised. A better explanation in terms of procedures followed for the development of both the questionnaire and the interviews must be described within the text.

Response 3: We agree with the reviewer. We have added more information in section 2, to ensure a better understanding of the readers. We added details on study design, and our logic to choose sites, outcomes, and such.

Point 4: 4) Section 4 must be revised. In its current state, it seems the sum of independent paragraphs.

Response 4: The discussion was revised to improve its flow and coherence. Also, a more straight-forward conclusion section was added 

Point 5: 5) No conclusions are reported. Please, consider a dedicated section. 

Response 5: We appreciate this point and we agree with the reviewer, we included a conclusion section as mentioned before

Reviewer 3 Report

Overall a good initial study, ostensibly the first of its kind in Chile (Latin America),  that indicates more research is needed to more fully assess health impacts.    

Minor text edits and clarification are needed as follows: 1. Introduction; a). 5th paragraph- 5th sentence- "has boost the economy" [boosted]..."because of this the has the highest" [take out the word "the"]. 

b) - 7th paragraph- "The purpose of this article is to describe the design and population characteristics of a ....." - is describing the "design and populations characteristics "really the purpose of the article?...need clarification or restating. 

3. Results- 2nd paragraph...states that ewaste recylcijng ws the main source of income for most participants (83%)...in 3rd paragraph it states that ...65% reported e-waste as  their main source of income...needs clarification or correction. 

4. Discussion; 7th paragraph- Supplementary Materials...add period at end of last sentence.

Author Response

Response to Reviewer 3 Comments

Overall a good initial study, ostensibly the first of its kind in Chile (Latin America), that indicates more research is needed to more fully assess health impacts.    

Point 1: Minor text edits and clarification are needed as follows: 1. Introduction; a). 5th paragraph- 5th sentence- "has boost the economy" [boosted]..."because of this the has the highest" [take out the word "the"].

Response 1: We appreciate the reviewer comment; we have corrected this comment.

Point 2: b) - 7th paragraph- "The purpose of this article is to describe the design and population characteristics of a ....." - is describing the "design and populations characteristics "really the purpose of the article?...need clarification or restating. 

Response 2: We considered this point on the title along with the overall point about the scope of the paper. We better defined the scope of the paper and, consequently, we rewrote several paragraphs of the introduction to give a clearer of the objective.

Point 3: 3. Results- 2nd paragraph...states that ewaste recylcijng ws the main source of income for most participants (83%)...in 3rd paragraph it states that ...65% reported e-waste as their main source of income...needs clarification or correction. 

Response 3: We agree with the reviewer. The question was asked twice (in the first the participants could answer more than 1 option and in the second there were excluding categories) the report was left for the first question (Table 2) for a better understanding of the results.

Point 4: 4. Discussion; 7th paragraph- Supplementary Materials...add period at end of last sentence.

Response 4: We appreciate the reviewer comment; we have corrected this comment.

Round  2

Reviewer 2 Report

Dear authors,

Thanks for having improved the work considering my requirements.

However, before publication some additional refinements are needed:

1) The English language must be revised by a mothertongue speaker;

2) Some tables are cutted into two parts without any logic. Please, revise them.

3) Please, follow the authors' guideline of the journal and re-structure th paper accordingly.

4) Lists of tables are putted within the text without a logic. Please revise them.

Best regards. 

Author Response

Dear reviewer,

We appreciate the comments sent in the first revision, which allowed us to make considerable improvements in our manuscript.

In relation to the current observations:

Point 1: The English language must be revised by a mothertongue speaker

Response 1: The writing was reviewed by two native English language authors.

Point 2: Some tables are cutted into two parts without any logic. Please, revise them.

Response 2: We have revised and corrected the tables so that they are not divided into two parts.

Point 3: Please, follow the authors' guideline of the journal and re-structure th paper accordingly.

Response 3: We have reviewed the guidelines for the authors and the format aspects have been modified in accordance with the guidelines.

Point 4: Lists of tables are putted within the text without a logic. Please revise them.

Response 4: We have corrected this error that occurred due to the format of the table titles.
